# The Compact Support Neural Network

**DOI:** 10.3390/s21248494

**Published:** 2021-12-20

**Authors:** Adrian Barbu, Hongyu Mou

**Affiliations:** Statistics Department, Florida State University, Tallahassee, FL 32306, USA; hm15f@my.fsu.edu

**Keywords:** neural networks, RBF networks, OOD detection, universal approximation

## Abstract

Neural networks are popular and useful in many fields, but they have the problem of giving high confidence responses for examples that are away from the training data. This makes the neural networks very confident in their prediction while making gross mistakes, thus limiting their reliability for safety-critical applications such as autonomous driving and space exploration, etc. This paper introduces a novel neuron generalization that has the standard dot-product-based neuron and the radial basis function (RBF) neuron as two extreme cases of a shape parameter. Using a rectified linear unit (ReLU) as the activation function results in a novel neuron that has compact support, which means its output is zero outside a bounded domain. To address the difficulties in training the proposed neural network, it introduces a novel training method that takes a pretrained standard neural network that is fine-tuned while gradually increasing the shape parameter to the desired value. The theoretical findings of the paper are bound on the gradient of the proposed neuron and proof that a neural network with such neurons has the universal approximation property. This means that the network can approximate any continuous and integrable function with an arbitrary degree of accuracy. The experimental findings on standard benchmark datasets show that the proposed approach has smaller test errors than the state-of-the-art competing methods and outperforms the competing methods in detecting out-of-distribution samples on two out of three datasets.

## 1. Introduction

Neural networks have proven to be extremely useful in many applications, including object detection, speech and handwriting recognition, and medical imaging, etc. They have become the state-of-the-art in these applications, and in some cases they even surpass human performance. However, neural networks have been observed to have a major disadvantage: they do not know when they do not know, i.e., do not know when the input is out-of-distribution (OOD), i.e., far away from the data they have been trained on. Instead of saying “I do not know”, they give some output with high confidence [1,2]. An explanation of why this is happening for the rectified linear unit (ReLU) based networks has been given in [3]. This issue is very important for safety-critical applications such as space exploration, autonomous driving, and medical diagnosis, etc. In these cases it is important that the system knows when the input data are outside its nominal range to alert the human (e.g., driver for autonomous driving or radiologist for medical diagnostic) to take charge in such cases.

A common way to address the problem of high confidence predictions for OOD examples is through ensembles [4], where multiple neural networks are trained with different random initializations, and their outputs are averaged in some way. The reason why ensemble methods have low confidence on OOD samples is that each NN’s high-confidence domain is random outside the training data, and the common high-confidence domain has therefore shrunk through averaging. This works well when the representation space (the space of the NN before the output layer) is high-dimensional but fails when this space is low-dimensional  [5].

Another popular approach is adversarial training [6], where the training is augmented with adversarial examples generated by maximizing the loss starting from perturbed examples. This method is modified in adversarial confidence enhanced training (ACET) [3], where the adversarial samples are added through a hybrid loss function. However, the instance space is extremely vast when it is high dimensional, and a finite number of training examples can only cover an insignificant part, and no matter how many OOD examples are used, there will always be parts of the instance space that have not been explored. Other methods include the estimation of the uncertainty using dropout [7], softmax calibration [8], and the detection of OOD inputs [9]. CutMix [10] is a method to generate training samples with larger variability, which helps improve generalization and OOD detection. All these methods are complementary to the proposed approach and could be used together with the classifiers introduced in this paper to improve accuracy and OOD detection.

In [11], two auto-regressive models are trained, one for the foreground in-distribution data and one for the background, and the likelihood ratio is used to decide for each observation whether it is OOD or not. This is a generative model, while our model is discriminative.

A number of works assume that the distance in the representation space is meaningful. A trust score was proposed in [12] to measure the agreement between a given classifier and a modified version of a *k*-nearest neighbor (*k*-NN) classifier. While this approach does consider the distance of the test samples to the training set, it only does so to a certain extent since the *k*-NN does not have a concept of “too far” and is also computationally expensive. A simple method based on the Mahalanobis distance is presented in [13]. It assumes that the observations are normally distributed in the representation space, with a shared covariance matrix for all classes. Our distribution assumption is much weaker, assuming that the observations are clustered into a number of clusters, not necessarily Gaussian. In our representation, each class is usually covered by one or more compact support neurons, and each neuron could be involved in multiple classes. Furthermore, ref. [13] simply replaces the last layer of the NN with their Mahalanobis measure and makes no attempt to further train the new model, while the CSN layers can be trained together with the whole network.

The generalized ODIN [14] decomposes the output into a ratio of a class-specific function $h_{i}\left(x\right)$ and a common denominator g(x), both defined over instances x of the representation space. Good results are obtained using $h_{i}$ based on the Euclidean distance or cosine similarity. Again, this approach assumes that the observations are grouped in a single cluster for each class, which explains why it uses very deep models (with 34–100 layers) that are more capable of obtaining representations that satisfy this assumption. Our method does not make the single cluster per class assumption and can use deep or shallow models.

The deterministic uncertainty quantification (DUQ) [5] method uses an RBF network and a special gradient penalty to decrease the prediction confidence away from the training examples. The authors also propose a centroid updating scheme to handle the difficulties in training an RBF network. They claim that regularization of the gradient is needed in deep networks to enforce a local Lipschitz condition on the prediction function that will limit how fast the output will change away from the training examples. While their smoothness and Lipschitz conditions might be necessary conditions, they are not sufficient conditions since a smoothly changing function could still have arbitrarily high confidence far away from the training examples. In contrast, our proposed compact support neural network (CSNN) is guaranteed to have zero outputs away from the training examples, which reflects in the lowest possible confidence. Furthermore, the maximum gradient of the CSN layer can be computed explicitly, and the Lipschitz condition can be directly enforced by decreasing the neuron support and weight decay. The authors of DUQ also encourage their gradient to be bounded away from zero everywhere, which they recognize is based on a speculative argument. In contrast, the CSNN gradient is zero away from the training examples, while still obtaining better OOD detection and smaller test errors than DUQ.

The contributions of this paper are the following:It introduces a novel neuron formulation that generalizes the standard neuron and the radial basis function (RBF) neuron as two extreme cases of a shape parameter. Moreover, one can smoothly transition from a regular neuron to an RBF neuron by gradually changing this parameter. The RBF correspondent to a ReLU neuron is also introduced and observed to have compact support, i.e., its output is zero outside a bounded domain.It introduces a novel way to train a compact support neural network (CSNN) or an RBF network, starting from a pre-trained regular neural network. For that purpose, the construction mentioned above is used to smoothly bend the decision boundary of the standard neurons, obtaining the compact support or RBF neurons.It proves the universal approximation theorem for the proposed neural network, which guarantees that the network will approximate any function from $L^{p}\left(R^{k}\right)$ with an arbitrary degree of accuracy.It shows through experiments on standard datasets that the proposed network usually outperforms existing out-of-distribution detection methods from the literature, both in terms of smaller test errors on the in-distribution data and larger Areas under the ROC curve (AUROC) for detecting OOD samples.

## 2. Materials and Methods

This paper investigates the neuron design as the root cause of high confidence predictions on OOD data and proposes a different type of neuron to address its limitations. The standard neuron is $f\left(x\right)=\sigma \left(w^{T}x+b\right)$, a projection (dot product) $x\to w^{T}x+b$ onto a direction w, followed by a nonlinearity σ(·). In this design, the neuron has a large response for vectors $x\in R^{p}$ that are in a half-space. This can be an advantage when training the neural network (NN) since it creates high connectivity in the weight space and makes the neurons sensitive to far-away signals. However, it can be a disadvantage when using the trained NN, since it leads to neurons unpredictably firing with high responses to far-away signals, which can result (with some probability) in high confidence responses of the whole network for examples that are far away from the training data.

To address these problems, a type of radial basis function (RBF) neuron [15], $f\left(x\right)=g(∥x-\mu {∥}^{2})$, is used and modified to have zero response at some distance *R* from μ. Therefore, the neuron has compact support, and the same applies to a layer formed entirely of such neurons. Using one such compact support layer before the output layer, one can guarantee that the space where the NN has a nonzero response is bounded, obtaining a more reliable neural network.

The loss function of such a compact support NN has many flat areas, and it can be difficult to train directly by backpropagation. However, the paper introduces a different way to train it, by starting with a trained regular NN and gradually bending the neuron decision boundaries to make them have smaller and smaller support.

The compact support neural network consists of a number of layers, where the layer before last contains only compact support neurons, which will be described next. The last layer is a regular linear layer without bias, so it can output an all-zero vector when appropriate.

### 2.1. The Compact Support Neuron

The radial basis function (RBF) neuron [15], $f\left(x,w\right)=g(∥x-w{∥}^{2})$, for $x,w\in R^{d}$, has a g(u)=exp(−βu) activation function. However, in this paper g(u)=max(R−u,0) will be used because it is related to the ReLU.

**A flexible representation.** Introducing an extra parameter α=1, the RBF neuron can be written as:(1)$$f_{\alpha }\left(x,w\right)={g(\alpha (∥x∥}^{2}+{∥w∥}^{2})-2w^{T}x).$$

Using the parameter α, one can smoothly interpolate between an RBF neuron when α=1 and a standard projection neuron when α=0. However, starting with an RBF neuron with g(u)=exp(−βu), the projection neuron is obtained for α=0 as $f_{\alpha }\left(x,w\right)=\exp\left(2w^{T}x\right)$ but which has an undesirable exponential activation function.

**The compact support neuron.** In order to obtain a standard ReLU based neuron $f_{\alpha }\left(x,w\right)=\rho \left(w^{T}x\right)$ with ρ(u)=max(u,0) for α=0, the activation g(u)=ρ(R−u) will be used, and the above construction is modified to obtain the compact support neuron:(2)$$f_{\alpha }\left(x,w,b,R\right)=\rho [\alpha (R-{∥x∥}^{2}-{∥w∥}^{2}-b)+2w^{T}x+b],$$
where a bias term *b* is also introduced for the standard neuron. Usually b=0 is used for simplicity.

The parameter *R* determines the radius of the support of the neuron when α>0. In fact, one can easily check that the support of $f_{\alpha }\left(x,w,b,R\right)$ from Equation (Equation 2) (i.e., the domain where it takes nonzero values) is a sphere of radius
(3)$$R_{\alpha }^{2}=R+b\left(1/\alpha -1\right)+{∥w∥}^{2}\left(1/\alpha^{2}-1\right)$$
centered at $w_{\alpha }=w/\alpha$. Therefore, the neuron from (Equation 2) has compact support for any α>0 and the support becomes tighter as α increases. Figure 1a shows the response on a 1D input *x* for the RBF neuron $y=\exp(-|x-2{|}^{2})$ and the compact support neuron $y=f_{\alpha }\left(x,2,0,1\right)$ from Equation (Equation 2) for α∈{0,0.8,1}. Observe that the standard neuron with ReLU activation y=max(x,0) is obtained as $y=f_{\alpha }\left(x,2,0,1\right)$ for α=0. Figure 1b shows on a 2D example the support of $f_{\alpha }\left(x,w,b,R\right)$ from (Equation 2) for several values of α∈[0,1], where $x=(x_{1},x_{2})$, w=(0,2), b=0 and R=1.

### 2.2. The Compact Support Neural Network

For a layer containing only compact support neurons (CSN), the weights can be combined into a matrix $W^{T}=\left(w_{1},\cdots ,w_{K}\right)$, the biases into a vector $b=(b_{1},\cdots ,b_{K})$ and the radii into a vector $r=(R_{1},\cdots ,R_{K})$ and the CSN layer can be written as: (4)$$f_{\alpha }\left(x,W,b,r\right)=\rho (\alpha \left[r-b-x^{T}x-\mathrm{Tr}\left(WW^{T}\right)\right]+2Wx+b),$$
where $f_{\alpha }\left(x,W,b,r\right)={\left(f_{1}\left(x\right),\cdots ,f_{K}\left(x\right)\right)}^{T}$ is the vector of neuron outputs of that layer. This formulation enables the use of standard neural network machinery (e.g., PyTorch) to train a CSN. In practice usually no bias term is used (i.e., b=0), except in low dimension experiments. The radius parameter r is trainable.

The simplest compact support neural network (CSNN) has two layers: a hidden layer containing compact support neurons (Equation 2) and an output layer, which is a standard fully connected layer without bias, as shown in Figure 2a. A Batch Normalization layer without trainable parameters can be added to normalize the data as described below. A LeNet network with a CSN layer before the last layer is shown in Figure 2b.

**Normalization.** It is desirable that ∥x∥ be approximately 1 on the training examples so that the value of the radius *R* does not depend on the dimension *d* of x. These goals can be achieved by standardizing the variables to have zero mean and standard deviation $1/\sqrt{d}$ on the training examples. This way ${∥x∥}^{2}∼1$ when the dimension *d* is large (under assumptions of normality and independence of the variables of x). Our experiments on three real datasets indicate that indeed ∥x∥∼1 when the inputs x are normalized this way. To achieve this goal, a Batch Normalization layer without trainable parameters is added before the CSN layer.

  **Training.** Similar to the RBF network, training a neural network with such neurons with α=1 is difficult because the loss function has many local optima. To make matters even worse, the compact support neurons have small support when α is close to 1, and consequently the loss function has flat regions between the local minima.

This is why another training approach is taken. Using Equation (Equation 4), a CSNN can be trained by first training a regular NN (α=0) and then gradually increasing the shape parameter α from 0 towards 1 while continuing to update the NN parameters. Observe that whenever α>0, the NN has compact support, but the support becomes smaller as α becomes closer to 1. The training procedure is described in detail in Algorithm 1.
**Algorithm 1** Compact Support Neural Network (CSNN) Training **Input:** Training set $T={\left\{\left(x_{i},y_{i}\right)\in R^{p}\times R\right\}}_{i=1}^{n}$,
 **Output:** Trained CSNN. 1:  Train a regular NN f(x)=Lρ(2Wg(x)+b) where W,L are the last two layer weight matrices and g(x) is the rest of the NN. 2:  Freeze g(x), compute $u_{i}=g\left(x_{i}\right),i=1,\cdots ,n$, their mean μ and standard deviation σ. 3:  Obtain normalized versions $v_{i}$ of $u_{i}$ as $v_{i}=\left(u_{i}-\mu \right)/\sqrt{d}\sigma ,i=1,\cdots ,n.$ 4:  **for** e= 1 to $N^{epochs}$ **do** 5:     Set $\alpha =e/N^{epochs}$ 6:     Use the examples $\left(v_{i},y_{i}\right)$ to update (W,L,b,r) based on one epoch of
$$f(v)=L\rho \left(\alpha \left[r-v^{T}v-\mathrm{Tr}\left(WW^{T}\right)-b\right]+2Wv+b\right)\;$$ 7:  **end for**

In practice, the training is stopped at an α≤1 where the training and validation errors still take acceptable values, e.g., a validation error less than the validation error for α=0. However, the larger the value of α, the tighter the support is around the training data and the better the generalization. The whole network can then be fine-tuned using a few more epochs of backpropagation.

### 2.3. Datasets Used

Real data experiments are conducted on classifiers trained on three datasets: MNIST [16], CIFAR-10 and CIFAR-100 [17]. The OOD (out of distribution) detection is evaluated using the respective test sets as in-sample data and other datasets as OOD data, such as the test sets of EMNIST [18], FashionMNIST [19] and SVHN [20], and the validation set of ImageNet [21]. For MNIST, a grayscale version of CIFAR-10 is also used as OOD data, obtained by converting the 10,000 test images to grayscale and resizing them to 28×28.

### 2.4. Neural Network Architecture

For MNIST, a 4-layer LeNet convolutional neural network (CNN) backbone is used, with two 5×5 convolution layers with 32 and 64 filters, respectively, followed by ReLU and 2×2 max pooling, and two fully connected layers with 256 and 10 neurons. For the other two datasets, a ResNet-18 [22] backbone is used, with 4 residual blocks with 64, 128, 256 and 512 filters, respectively.

For the CSNN, two architectures, illustrated in Figure 2, will be investigated. The first is a small one (called CSNN), illustrated in Figure 2a, which takes the output of the last convolutional layer of the backbone as input, normalized as described in Section 2.2 using a batch normalization layer without any learnable affine parameters. The second one is a full network (called CSNN-F), illustrated in Figure 2b, where the backbone (LeNet or ResNet) is part of the backpropagation, and a Batch Normalization layer (BN) without any learnable parameters is used between the backbone and the CSN layer.

All experiments were conducted on an MSI GS-60 Core I7 laptop with 16GB RAM and Nvidia GTX 970M GPU, running the Windows 10 operating system. The CSNN and CSNN-F networks were implemented in PyTorch 1.90.

### 2.5. Training Details

For all datasets, data augmentation with padding (3 pixels for MNIST, 4 pixels for the rest) and random cropping is used to train the backbones. For CIFAR-100, random rotation up to 15 degrees is also used.No data augmentation is used when training the CSNN and CSNN-F.

The CSNN was trained for 510 epochs with R=0.01, of which 10 epochs at α=0. The Adam optimizer is used with a learning rate of 0.001 and weight decay of 0.0001. SGD obtained similar results. The CSNN-F was trained with SGD with a learning rate of 0.001 and weight decay of 0.0005. Its layers were initialized with the trained backbone and the trained CSNN. Then α was kept fixed for two epochs and increased by 0.005 every epoch for 4 more epochs.

Training the CSNN from α=0 to α=1 for 510 epochs takes less than an hour. Each epoch of the CSNN-F takes less than a minute with the LeNet backbone and about 3 min with the ResNet-18 backbone. The CSNN-F was obtained by merging the corresponding CSNN head with the ResNet or LeNet backbone and training them together for 6 epochs.

### 2.6. Ood Detection

The out of distribution (OOD) detection is performed similarly to the way it is performed in a standard CNN. For any observation, the maximum value of the network’s raw outputs is used as the OOD score for predicting whether the observation is OOD or not. If the observation is in-distribution, its score will usually be large, and if it is OOD, it will usually be close to zero or even zero. The ROC (receiver operating characteristic) curve based on these scores for the test set of the in-distribution data (as class 0) and one OOD dataset (as class 1) will give us the AUROC (Area under the ROC). If the two distributions are not separable (have concept overlap), some of the OOD scores will be large, but for the OOD observations that are away from the area of overlap they will be small or even zero.

### 2.7. Methods Compared

The OOD (out of distribution) detection results of the CSNN and CSNN-F are compared with the Adversarial Confidence Enhanced Training (ACET) [3], Deterministic Uncertainty Quantification (DUQ) [5], a standard CNN and a standard RBF network. The RBF network has the same architecture as the CSNN-F, but with an RBF layer instead of the CSN layer, and has a learnable σ for each neuron. Also shown are results for an ensemble of five or 10 CNNs trained with different random initializations; however, these methods are more computationally expensive and are not included in our comparison. The ACET results are taken directly from [3], and the DUQ, CNN and ensemble results were obtained using the DUQ authors’ PyTorch code. The parameters for the CNN, RBF and ensemble were tuned to obtain the smallest average test error. For DUQ, multiple models were trained with various combinations of the length scales σ∈{0.05,0.1,0.2,0.3,0.5,1.0} and gradient penalty λ∈{0,0.05,0.1,0.2,0.3,0.5,1.0} and the combination with the best test error-AUROC trade-off was selected. For the CSNN methods, for each dataset, the classifier was selected to correspond to the largest α where the test error takes a value comparable to the other methods compared.

## 3. Results

This section presents theoretical results on the gradient of the proposed CSNN and its universal approximation properties, plus experimental results on a number of public datasets.

### 3.1. Gradient Bound

The authors of the Deterministic Uncertainty Quantification (DUQ) [5] paper claim that gradient regularization is needed in deep networks to enforce a local Lipschitz condition on the prediction function that limits how fast the output will change away from the training examples. Thus, it is of interest to see whether this Lipschitz condition is satisfied for the proposed CSNN.

The following result proves that indeed the Lipschitz condition [5] is satisfied for the CSN layer:

**Theorem** **1.**
*The gradient of a CSN neuron is bounded by:*

(5)
$$∥\nabla_{x}f_{\alpha }{\left(x,w,b,R\right)∥}^{2}\leq \alpha^{2}R+b\alpha \left(1-\alpha \right)+{∥w∥}^{2}\left(1-\alpha^{2}\right)$$



**Proof.** The maximum gradient of a CSN neuron can be explicitly computed as follows. The gradient of
$$f_{\alpha }\left(x,w,b,R\right)=\rho \left[\alpha \left(R-x^{T}x-w^{T}w-b\right)+2w^{T}x+b\right],$$
where ρ(u)=max(u,0), is:
$$\nabla_{x}f_{\alpha }\left(x,w,b,R\right)=-\alpha x+w\;\mathrm{if}\;\alpha \left(R-x^{T}x-w^{T}w-b\right)+2w^{T}x+b\geq 0,\;\mathrm{otherwise}\;0.$$The above inequality can be rearranged after dividing by α and regrouping terms as:
$$x^{T}x-2w^{T}x/\alpha +w^{T}w/\alpha^{2}\leq R-b+b/\alpha -w^{T}w+w^{T}w/\alpha^{2}=R_{\alpha }^{2},$$
where $R_{\alpha }^{2}$ was defined in Equation (Equation 3). Thus, the gradient is
(6)$$\nabla_{x}f_{\alpha }\left(x,w,b,R\right)=-\alpha x+w\;\mathrm{if}\;{∥x-w/\alpha ∥}^{2}\leq R_{\alpha }^{2},\;\mathrm{otherwise}\;0,$$
and therefore, the maximum gradient norm is:
$$\max_{x}∥\nabla_{x}f_{\alpha }{\left(x,w,b,R\right)∥}^{2}\;=\;\max_{x}\alpha^{2}{∥x\;-\;w/\alpha ∥}^{2}\;=\;\alpha^{2}R_{\alpha }^{2}\;=\;\alpha^{2}R+b\alpha \left(1\;-\;\alpha \right)+{∥w∥}^{2}\left(1\;-\;\alpha^{2}\right),$$
where the last equality was again obtained using Equation (Equation 3). □

Thus, a small gradient can be enforced everywhere by penalizing the *R* and ${∥w∥}^{2}$ to be small using weight decay, or by making α close to 1, or both (all assuming b=0).

### 3.2. Universal Approximation for the Compact Support Neural Network

It was proved in [23,24] that standard two-layered neural networks have the universal approximation property, in the sense that they can approximate continuous functions or integrable functions with arbitrary accuracy under certain assumptions. Similar results have been proved for RBF networks in [25,26]; thus, it is of interest whether such results can be proved for the proposed compact support neural network. In this section, the universal approximation property of the compact support neural networks will be proved.

Let $L^{\infty }\left(R^{d}\right)$ and $L^{p}\left(R^{d}\right)$ and be space of functions $f:R^{d}\to R$ such that they are essentially bounded and, respectively, their *p*-th power $f^{p}$ is integrable. Denote the $L^{p}$ and $L^{\infty }$ norms by $\left|\right|-{\left|\right|}_{p}$ and $\left|\right|-{\left|\right|}_{\infty }$, respectively.

The family of networks considered in this study consists of two-layer neural networks, which can be written as:(7)$$q\left(x\right)=\sum_{i=1}^{H}\beta_{i}f\left(x,w_{i},b_{i}\right)$$
where *H* is the number of hidden nodes, $\beta_{i}\in R$ are the weights from the *i*-th hidden node to the output nodes, $f(x,w_{i},b_{i})$ is the representation of the hidden neuron with weights $w_{i}\in R^{n}$ and biases $b_{i}\in R$.

The neurons used in the hidden layer can be either regular neurons or radial-basis function (RBF) neurons. The regular neuron can be written as:(8)$$f\left(x,w,b\right)=g\left(w^{T}x+b\right)$$
where *g* is an activation function such as the sigmoid or ReLU. An RBF neuron has the following representation:(9)$$f\left(x,w,b\right)=g\left(\frac{\left||x-w|\right|}{b}\right)$$
where the exponential function g(x)=exp(−x) is used for activation in RBF networks.

The studies on regular neurons [23,24] showed that if the activation function *g* used in the hidden layer is continuous almost everywhere, locally essentially bounded, and not a polynomial, then a two-layered neural network can approximate any continuous function with respect to the $\left|\right|-{\left|\right|}_{\infty }$ norm.

Universal approximation results for RBF networks are quite limited according to [25,26], which proved that if the RBF neuron used in the hidden layer is continuous almost everywhere, bounded and integrable on $R^{n}$, the RBF network can approximate any function in $L^{p}\left(R^{n}\right)$ with respect to the $L^{p}$ norm with 1≤p≤∞. More exactly, in [25], the following statement is proved:

**Theorem** **2.**
*(Park 1991) Let $K:R^{d}\to R$ be an integrable bounded function such that K is continuous almost everywhere and $\int_{R^{d}}K\left(x\right)dx\neq 0$. Then the family of functions $q\left(x\right)=\sum_{i=1}^{H}\beta_{i}K\left(\frac{x-z_{i}}{\sigma }\right)$ is dense in $L^{p}\left(R^{d}\right)$ for every p∈[1,∞).*


The above result will be used to prove the following statement for the CSNN:

**Theorem** **3.**
*Let α∈(0,1], $R_{0}\geq 0$, and let g:R→R be any non-negative, continuous increasing activation function. Then the family of two-layer compact support neural networks: $q\left(x\right)=\sum_{i=1}^{H}\beta_{i}f_{\alpha }\left(x,w_{i},R_{w_{i}}\right)$, with $f_{\alpha }\left(x,w,R\right)=g(\alpha (R-{∥x∥}^{2}-{∥w∥}^{2})+2w^{T}x)$, and*

$$R_{w}=\frac{1}{\alpha }R_{0}-{∥w∥}^{2}\left(\frac{1}{\alpha^{2}}-1\right)$$

*is dense in $L^{p}\left(R^{d}\right)$ for every p∈[1,∞).*


**Proof.** Since 0<α≤1, it implies that:
(10)$$\begin{array}{cc} f_{\alpha }\left(x,w,R\right) & =g(\alpha (R\;-\;{∥x∥}^{2}\;-\;{∥w∥}^{2})+2w^{T}x)\\ \; & =g(\alpha (R\;-{\;∥w∥}^{2}+\frac{{∥w∥}^{2}}{\alpha^{2}})-\alpha \left(x^{T}x-2\frac{w^{T}}{\alpha }x+\frac{w^{T}w}{\alpha^{2}}\right))\\ \; & =g[\alpha (R\;+{\;∥w∥}^{2}\left(\frac{1}{\alpha^{2}}\;-\;1\right))\;-\;\alpha ∥x\;-\;\frac{w}{\alpha }{∥}^{2}]. \end{array}$$Therefore,
$$f_{\alpha }\left(x,w,R_{w}\right)=K\left(\frac{x-w/\alpha }{\sigma }\right)$$
where $\sigma =1/\sqrt{\alpha }$, $K\left(x\right)=g(R_{0}-∥x{∥}^{2})$ and
$$R_{w}=\frac{1}{\alpha }R_{0}-{∥w∥}^{2}\left(\frac{1}{\alpha^{2}}-1\right).$$Observe that K(x) is bounded because g≥0 is increasing and thus
$$0\leq K\left(x\right)=g(R_{0}-{∥x∥}^{2})\leq g\left(R_{0}\right).$$Since K(x) is also integrable and continuous and $\int_{R^{d}}K\left(x\right)dx\neq 0$, Theorem 2 applies with $z_{i}=w_{i}/\alpha$ and $\sigma =1/\sqrt{\alpha }$, obtaining the desired result. □

Observe that universal approximation results for α=0 have already been proven in [23,24] and that the standard RBF kernel can be obtained in Theorem 3 by taking g(x)=exp(x), α=1 and $R_{0}=0$.

### 3.3. Two-Dimensional Example

This experiment is on the moons 2D dataset, where the data are organized on two intertwining half-circle-like shapes, one containing the positives and one the negatives. The data are scaled so that all observations are in the interval ${\left[0,1\right]}^{2}$ (shown as a white rectangle in Figure 3. The out of distribution (OOD) data are generated starting with 100×100= 10,000 samples on a grid spanning ${\left[-0.5,1.5\right]}^{2}$ and removing all samples at a distance at most 0.1 from the moons data, obtaining 8763 samples.

A two-layer CSNN is used, with 128 CSN neurons in the hidden layer, as illustrated in Figure 2a. The CSNN is trained on 200 training examples for 2000 epochs with R=0.02 and α increasing from 0 to 1 as
$$\alpha_{i}=\min\left(1,\max\left(0,\left(i-100\right)/1500\right)\right),i=1,\cdots ,2000.$$

The confidence map for the trained classifier is shown in Figure 3d. One can see that the confidence is 0.5 (white) almost everywhere except near the training data, where it is close to 1 (black). This assures us that the method works as expected, shrinking the support of the neurons to a small domain around the training data. One can also see that the support is already reasonably small for α=0.6, and it becomes tighter and tighter as α becomes closer to 1.

The training/test errors vs. α are shown in Figure 4. Also shown is the AUROC (area under the ROC curve) for detecting the OOD data described above against the test set. Observe that the training and test errors for α=0 are quite large because the standard 2-layer NN with 128 neurons cannot fit these data well enough and decrease as the neuron support decreases in which the model is better capable of fitting the data.

It is known [3,27] that the output of a ReLU-based neural network is piecewise linear, and the domains of linearity are given by the activation pattern of the neurons. The activation pattern of the neurons contains the domains where the set of active neurons (with nonzero output) does not change. These activation pattern domains are polytopes, as shown in Figure 5a, for a two-layer NN with 32 neurons. The activation domains for a CSNN are intersections of circles, as illustrated in Figure 5b, with the domain where all neurons are inactive shown in white. The corresponding confidence map is shown in Figure 5c.

In real data applications, one does not need to go all the way to α=1 since even for smaller α, the support is still bounded and if the instance space is high dimensional (e.g., 512 to 1024 in the real data experiments below), the volume of the support of the CNN will be extremely small compared to the instance space, making it unlikely to have high confidence on out-of-distribution data.

### 3.4. Real Data Experiments

In Figure 6, the train/test errors vs. α for the CSNN on the three datasets are shown. Also shown are the area under the ROC curve (AUROC) for OOD detection on CIFAR-10 or CIFAR-100. Observe that all curves on the real data are very smooth, even though they are obtained from one run, not averaged. One could see that the training and test errors stay flat for a while and then start increasing from a certain α, which depends on the dataset. At the same time, the AUROC stays flat and slightly increases, and there is a range of values of α where the test error is low and the AUROC is large.

Different values of α make different trade-offs between test error and AUROC. In practice, α should be chosen as large as possible where an acceptable validation error is still obtained to have the smallest support possible. For example, one could choose the largest α such that the validation error at α is less than or equal to the validation error at α=0.

The results are shown in Table 1. All results except the ACET results are averaged over 10 runs, and the standard deviation is shown in parentheses. The results of the best non-ensemble method are shown in bold and the second best in red.

## 4. Discussion

From Table 1, one could see that the trained RBF has difficulties in fitting the in distribution data well, with much larger test errors on CIFAR-10 and CIFAR-100 than the standard CNN. The most probable cause is that the training is stuck in a local optimum, because the training errors were also observed to be large (not shown in the table).

Further comparing the test errors, Table 1 reveals that the proposed CSNN-F method obtains the smallest test errors on in-distribution data among the non-ensemble methods compared. These findings strengthen the claim that the CSNN can fit the in-distribution data as well as a standard CNN, a claim that is also supported by the universal approximation statement from Theorem 3.

Comparing the detection AUROC on different OOD datasets, the proposed CSNN and CSNN-F methods obtain the best results on MNIST and CIFAR-100, and ACET and SNGP obtain the best results on CIFAR-10. However, the test errors of ACET are the highest among all methods by a large margin.

One should be aware that there usually is a trade-off to be made between small test errors and good OOD detection. ACET makes this trade-off in favor of a high AUROC by training with adversarial samples, while the other methods are trying to obtain small test errors, comparable with a standard CNN. Observe that the test errors of the CSNN-F approach are smaller than the CSNN, and the AUROCs are comparable.

Compared to ACET, both CSNN and CSNN-F obtain smaller test errors on all three datasets and better average AUROC on two out of three datasets. Compared to DUQ, the CSNN and CSNN-F obtain comparable test errors and better average AUROC on all three datasets. The RBF network cannot obtain a small training or test error in most cases, and the test errors and OOD detection results are poor and have a large variance.

Comparing the training time, the CSNN methods are about four times faster than training a 5-ensemble, eight times faster than a 10-ensemble and about three times faster than DUQ.

Overall, compared to the other non-ensemble OOD detection methods evaluated, the proposed methods obtain smaller test errors and better OOD detection performance (except ACET, which has large test errors). Training the whole deep network together with the CSNN results in smaller test errors and improved OOD detection performance.

In the synthetic experiment in Figure 3, the train and test errors were close to 0 for α=1 because there are neurons close to all observations. However, in the real data applications where the representation space is high dimensional, the training, test and validation errors might first decrease a little bit but ultimately increase as α approaches 1. For example, one could see the test errors vs. α for the synthetic dataset in Figure 4 and for the real datasets in Figure 6. This is due to the curse of dimensionality, which makes all distances between observations relatively large and the CSNN centers will also be far from the observations, thus the neurons will have a larger radius to cover the training data.

It is worth noting that in contrast to the weights of a standard neuron, the weights of the compact support neuron exist in the same space as the neuron inputs, and they can be regarded as templates. Thus, they have more meaning, and one could easily visualize the type of responses that make them maximal, using standard neuron visualization techniques such as [28]. Furthermore, one can also obtain samples from the compact support neurons, e.g., for generative or GAN models.

## 5. Conclusions

This paper presented four contributions. First, it introduced a novel neuron formulation that is a generalization of the standard projection-based neuron and the RBF neuron as two extreme cases of a shape parameter α∈[0,1]. It obtains a novel type of neuron with compact support by using ReLU as the activation function. Second, it introduced a novel way to train the compact support neural network of an RBF network by starting with a pretrained standard neural network and gradually increasing the shape parameter α. This training approach avoids the difficulties in training the compact support NN and the RBF networks, which have many local optima in their loss functions. Third, it proves the universal approximation property of the proposed neural network, in that it can approximate any function from $L^{p}\left(R^{d}\right)$ with arbitrary accuracy, for any p≥1. Finally, through experiments, it shows that the proposed compact support neural network outperforms the standard NN and the RBF network and even usually outperforms existing state-of-the-art OOD detection methods, both in terms of smaller test errors on in-distribution data and larger AUROC for detecting OOD samples.

The OOD detection feature is important in safety critical applications such as autonomous driving, space exploration and medical imaging.

The results have been obtained without any adversarial training or ensembling, and adversarial training or ensembling could be used in the proposed framework to obtain further improvements.

In real data applications, the compact support layer was used as the last layer before the output layer. This ensures that the compact support is involved in the most relevant representation space of the CNN. However, because the CNN still has many projection-based layers to obtain this representation, it means that the corresponding representation in the original image space does not have compact support, and erroneous high-confidence predictions are still possible. Architectures with multiple compact support layers that have even smaller support in the image space are subject to future study.

## Figures and Tables

**Figure 1 sensors-21-08494-f001:**
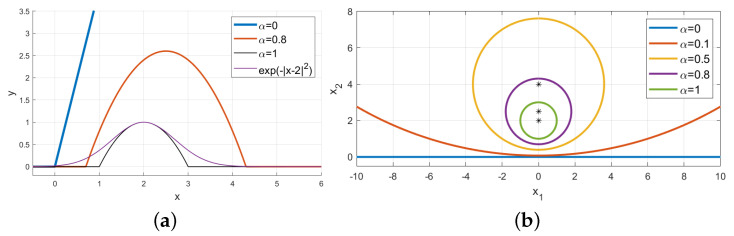
(**a**) 1D example. Comparison between RBF neuron $y=\exp(-|x-2{|}^{2})$ and compact support neurons $y=f_{\alpha }\left(x,2,0,1\right)$ from (Equation 2) for α∈{0,0.8,1}. (**b**) 2D example. The construction (Equation 2) smoothly interpolates between a standard neuron (α=0) and an RBF-type of neuron (α=1). Shown are the decision boundaries for $f_{\alpha }\left(x,w,0,1\right)$ with $x=(x_{1},x_{2})$, w=(0,2) for α∈{0,0.1,0.5,0.8,1} and the corresponding centers w/α as “*”.

**Figure 2 sensors-21-08494-f002:**
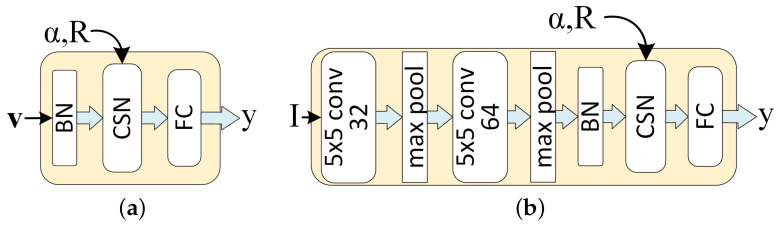
(**a**) A simple compact support neural network (CSNN), with the CSN layer described in (Equation 4). (**b**) A CSNN-F with LeNet backbone, where all layers are trainable.

**Figure 3 sensors-21-08494-f003:**
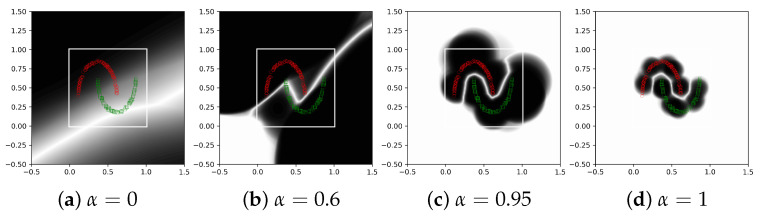
The confidence map (with white for 0.5 and black for 1) of the trained CSNN on the moons dataset for different values of α∈[0,1].

**Figure 4 sensors-21-08494-f004:**
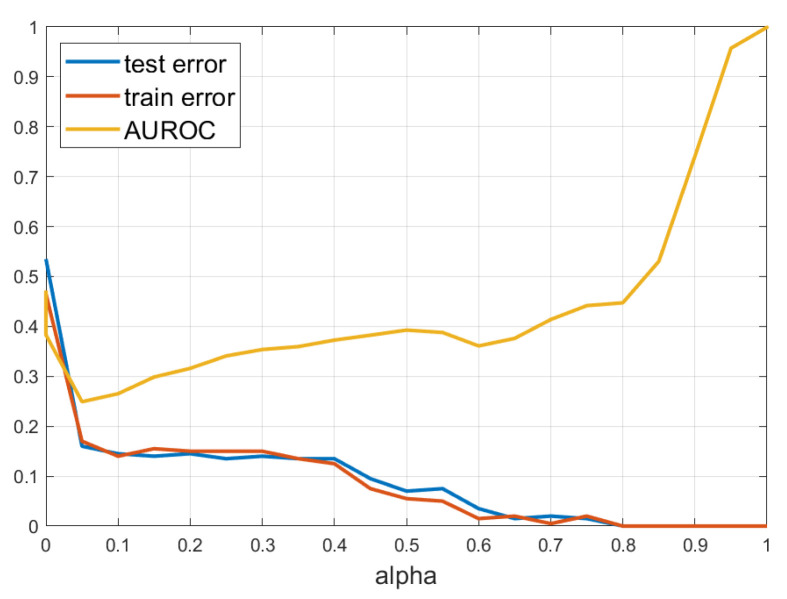
CSNN train and test errors, and AUROC for OOD detection vs. α for the moon data.

**Figure 5 sensors-21-08494-f005:**
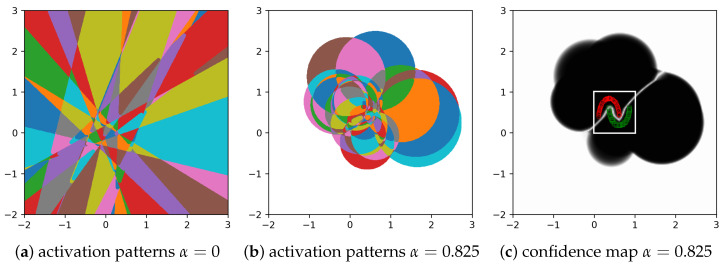
Example of activation pattern domains for a regular NN and a CSNN (α=0.825), and the resulting confidence map (white for 0.5 and black for 1) for α=0.825 for a 32 neuron 2-layer CSNN.

**Figure 6 sensors-21-08494-f006:**
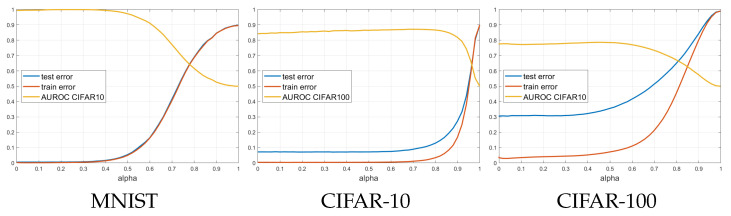
Train and test errors, and Area under ROC Curve (AUROC) for OOD detection vs. α for CSNN classifiers trained on three real datasets. These plots are obtained from one training run.

**Table 1 sensors-21-08494-t001:** OOD detection comparison in terms of area under the ROC curve (AUROC) for models trained and tested on several datasets. Best non-ensemble results are **bold** and second best are red. For each model, the test error in % is shown in the “Train on” row. The ACET results are taken from [3]. All other results are averaged over 10 runs, and the standard deviation is shown in parentheses.

	CNN	RBF	ACET	DUQ	SNGP	CSNN	CSNN-F	5 Ens	10 Ens
Trn. MNIST	0.53% (0.05)	0.65 (0.07)	0.66	0.57 (0.04)	0.54 (0.15)	0.52 (0.01)	**0.50** (0.02)	0.50 (0.02)	0.51 (0.03)
EMNIST	0.983 (0.001)	0.986 (0.004)	0.912	0.988 (0.001)	0.978 (0.007)	**0.992** (0.001)	0.990 (0.002)	0.985 (0.001)	0.985 (0.001)
FashMNIST	0.989 (0.001)	0.998 (0.002)	**0.998**	**0.998** (0.001)	0.988 (0.017)	**0.998** (0.001)	0.997 (0.001)	0.992 (0.001)	0.992 (0.001)
gryCIFAR-10	0.995 (0.001)	0.899 (0.040)	**1.000**	0.978 (0.005)	0.996 (0.002)	**1.00** (0.0001)	**1.00** (0.0001)	0.992 (0.002)	0.992 (0.002)
Average	0.989 (0.005)	0.961 (0.050)	0.97	0.988 (0.009)	0.987 (0.013)	**0.996** (0.003)	**0.996** (0.004)	0.990 (0.004)	0.990 (0.004)
Tr. CIFAR100	26.18% (0.28)	50.23 (1.8)	32.24	31.60 (0.44)	26.30 (0.26)	30.89 (0.10)	**24.46** (0.12)	22.43 (0.20)	21.86 (0.11)
CIFAR-10	0.750 (0.002)	0.652 (0.018)	0.72	0.719 (0.006)	0.739 (0.002)	**0.783** (0.001)	**0.762** (0.002)	0.781 (0.001)	0.786 (0.001)
SVHN	0.781 (0.035)	0.684 (0.071)	**0.912**	0.752 (0.043)	0.800 (0.009)	0.872 (0.001)	0.860 (0.006)	0.832 (0.013)	0.834 (0.009)
ImageNet	0.766 (0.002)	0.679 (0.012)	0.752	0.741 (0.004)	0.763 (0.001)	0.755 (0.001)	**0.793** (0.001)	0.798 (0.001)	0.803 (0.001)
Average	0.766 (0.023)	0.671 (0.045)	0.795	0.737 (0.028)	0.768 (0.026)	0.804 (0.050)	**0.805** (0.042)	0.804 (0.022)	0.808 (0.021)
Tr. CIFAR-10	**5.99%** (0.09)	18.40 (5.0)	8.44	6.73 (0.32)	6.35 (0.08)	7.28 (0.06)	6.18 (0.09)	4.83 (0.16)	4.59 (0.08)
CIFAR-100	0.860 (0.001)	0.765 (0.03)	0.852	0.838 (0.009)	**0.885** (0.001)	0.865 (0.001)	0.882 (0.003)	0.891 (0.001)	0.897 (0.001)
SVHN	0.899 (0.012)	0.810 (0.04)	**0.981**	0.914 (0.022)	0.923 (0.012)	0.908 (0.001)	0.900 (0.013)	0.917 (0.006)	0.924 (0.002)
ImageNet	0.834 (0.002)	0.755 (0.03)	0.859	0.824 (0.009)	**0.864** (0.001)	0.848 (0.001)	0.854 (0.004)	0.863 (0.001)	0.869 (0.001)
Average	0.865 (0.029)	0.777 (0.04)	**0.897**	0.859 (0.043)	0.891 (0.026)	0.874 (0.026)	0.879 (0.021)	0.890 (0.023)	0.897 (0.023)

## Data Availability

The following publicly available datasets have been used in experiments: MNIST [16], CIFAR-10 and CIFAR-100 [17], EMNIST [18], FashionMNIST [19], SVHN [20], and the validation set of ImageNet [21].

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
