# Peer review of "The Compact Support Neural Network"

_sensors, 2021, doi:10.3390/s21248494_

Round 1
Reviewer 1 Report
The paper proposes a compact support neural network, which deals with the out of distribution samples prudently. The method, performance and theory are well presented. Therefore, I am glad to suggest accepting this paper.
Author Response
We thank Reviewer 1 for the positive review.
Reviewer 2 Report
Pls find my review comments;
1. x and y axes are undefined!!! Infact results are not properly explained in detail.
2. There is many abbreviations that are not defining in their 1st appearence
For example: RBF neuron, ReLU etc.
3. In the section `Introduction':
2nd and 3rd paragraphs should be located there in section `2. Materials and Methods' in place of introduction.
4. From (6) to $\alpha^2 R_{\alpha}^2$ is ok, but thereafter how does final expression come direct as (5).
5. Theorem 2 with [25] does not make any sense, as this property has not been used anywhere in the paper.
Therefore, consideration of J. Park et al.'s theorem not clear to me?
6. Final expression of (10) is written directly, needs more steps!!!!
7. Theorems 1 and 3 need more steps, clarity, description and explanation to get into them.
8. Abstract and major contribution to be re-written. Moreover, contribution and novelty of the paper are not clear to me.
9. Introduction needs to be revised.
Author Response
- x and y axes are undefined!!! In fact results are not properly explained in detail.
Reply: We clarified in Figure 1 what the axes are both in the figure, the figure caption and the text referring to the figure.
- There is many abbreviations that are not defining in their 1st appearance. For example: RBF neuron, ReLU etc.
Reply: We added definitions for RBF, ReLU, NN, CNN, k-NN, ROC, AUROC
- In the section `Introduction': 2nd and 3rd paragraphs should be located there in section `2. Materials and Methods' in place of introduction.
Reply: We moved them to Section 2.
- From (6) to $\alpha^2 R_{\alpha}^2$ is ok, but thereafter how does final expression come direct as (5).
Reply: We clarified that it comes from eq. (3).
- Theorem 2 with [25] does not make any sense, as this property has not been used anywhere in the paper. Therefore, consideration of J. Park et al.'s theorem not clear to me?
Reply: Theorem 2 has been used in the last paragraph of the proof of theorem 3.
- Final expression of (10) is written directly, needs more steps!!!!
Reply: We added one more step to clarify
- Theorems 1 and 3 need more steps, clarity, description and explanation to get into them.
Reply: We added paragraphs placing the context for Theorems 1 and 3 and more steps for Theorems 1 and 3 to improve clarity.
- Abstract and major contribution to be re-written. Moreover, contribution and novelty of the paper are not clear to me.
Reply: We rewrote the abstract and major contribution to make more clear the contributions of the paper.
- Introduction needs to be revised.
Reply: We revised the introduction to also address Reviewer 3’s comments.
Reviewer 3 Report
The comments and suggestions are enclosed

Author Response
The authors presented their study on the Compact Support Neural Network and tried to present a neuron generalization that has the standard dot-product-based neuron and the RBF neuron as two extreme cases of a shape parameter. However, the following comments need to be addressed.
- The authors used words like "we presented," "we proved," and "we concluded" in the manuscript. The use of these words (I, we) is not the standard term to use in the articles.
Reply: We replaced all occurrences with the passive voice.
- The abstract should be refined by indicating the main findings of the presented study.
Reply: We reworked the abstract to clearly highlight the theoretical and experimental findings.
- The introduction part is very short; the authors directly describe their findings in the second paragraph. The first paragraph should be a clear introduction about the neural network, while the second and third paragraphs must contain relevant information about the study described in the literature, and the last paragraph should be the aim of the study, indicating the novelty of the presented study. It seems a discussion part.
Reply: We moved the literature review to the second to sixth paragraphs and refined the paper’s contributions to make the novelty more clear.
- In section 2, authors should provide clear information about the type of programming used and the software/system used for the employed functions.
Reply: We added a paragraph in Section 2.4 detailing the hardware, operating system and software sued in experiments.
- The explanation of Figure 1 is missing. The authors must explain the changes
provided.
Reply: We refined the Figure 1 caption and added a more detailed paragraph about Figure 1.
- Because there is limited information available with the relevant study, the authors must include more information in the discussion section.
Reply: We added more information in Section 4, separating the discussion about the test errors from the discussion about the OOD detection.
- The conclusion part is also very short; it should be clearer by indicating the main findings.
Reply: We rewrote the conclusion to make it clear what the main contributions and the main findings are.
Round 2
Reviewer 2 Report
Authors have well addressed all my previous round review comments. Therefore no more comments, the present form of the paper
is now ready for acceptance and publication.
Reviewer 3 Report
N/A